# Alterations in Circulating miRNA Levels after Infection with SARS-CoV-2 Could Contribute to the Development of Cardiovascular Diseases: What We Know So Far

**DOI:** 10.3390/ijms24032380

**Published:** 2023-01-25

**Authors:** Myrtani Pieri, Panayiotis Vayianos, Vicky Nicolaidou, Kyriacos Felekkis, Christos Papaneophytou

**Affiliations:** 1Department of Life Sciences, School of Life and Health Sciences, University of Nicosia, 2417 Nicosia, Cyprus; 2Non-Coding RNA Research Laboratory, School of Life and Health Sciences, University of Nicosia, 2417 Nicosia, Cyprus

**Keywords:** cardiovascular diseases, circulating miRNAs, SARS-CoV-2, COVID-19, cytokine storm, inflammation

## Abstract

The novel coronavirus disease 2019 (COVID-19) is caused by the severe acute respiratory syndrome coronavirus 2 (SARS-CoV-2) and poses significant complications for cardiovascular disease (CVD) patients. MicroRNAs (miRNAs) are small non-coding RNAs that regulate gene expression and influence several physiological and pathological processes, including CVD. This critical review aims to expand upon the current literature concerning miRNA deregulation during the SARS-CoV-2 infection, focusing on cardio-specific miRNAs and their association with various CVDs, including cardiac remodeling, arrhythmias, and atherosclerosis after SARS-CoV-2 infection. Despite the scarcity of research in this area, our findings suggest that changes in the expression levels of particular COVID-19-related miRNAs, including miR-146a, miR-27/miR-27a-5p, miR-451, miR-486-5p, miR-21, miR-155, and miR-133a, may be linked to CVDs. While our analysis did not conclusively determine the impact of SARS-CoV-2 infection on the profile and/or expression levels of cardiac-specific miRNAs, we proposed a potential mechanism by which the miRNAs mentioned above may contribute to the development of these two pathologies. Further research on the relationship between SARS-CoV-2, CVDs, and microRNAs will significantly enhance our understanding of this connection and may lead to the use of these miRNAs as biomarkers or therapeutic targets for both pathologies.

## 1. Introduction

Coronavirus disease-19 (COVID-19) is an emerging infectious disease caused by the widespread transmission of the severe acute respiratory syndrome coronavirus 2 (SARS-CoV-2), which threatens human health and public safety [1]. The outbreak of SARS-CoV-2 infection worldwide was declared a public health emergency by the World Health Organization (WHO) as of February 2020 [2]. SARS-CoV-2 is transmitted via contaminated aerosol droplets during close unprotected contact between infected and uninfected people. The virus infects humans through the mucosal membranes of the nose, mouth, and eyes [3].

Infection of the host cells is facilitated by the interaction of the viral spike protein, a surface protein, with the angiotensin-converting enzyme 2 (ACE2) integral protein that is expressed on the surface of the host cells (reviewed in [4]). The virus enters the host’s cells through receptor-mediated endocytosis (see [5] and references cited therein). The ACE2 protein is expressed on the surface of epithelial cells, including the alveolar lung cells [6]. Additionally, ACE2 is present in the endothelial cells of veins and arteries, the surface of enterocytes of the small intestine, and the arterial smooth muscle cells [7]. It has been proposed that this expression pattern increases the susceptibility of specific tissues to SARS-CoV-2 and, therefore, aids in understanding how multiple tissues/organs could be affected by the virus [8].

Certain underlying health conditions have been identified as high-risk factors for COVID-19, leading to higher mortality rates among COVID-19 patients with these conditions compared to those without [9]. Clinical studies have demonstrated an association between COVID-19 and cardiovascular disease (CVD). Furthermore, pre-existing CVD has been associated with worse outcomes and increased risk of death in patients with COVID-19 (reviewed in [10]). Specifically, early clinical results suggest that both the susceptibility to and the outcomes of COVID-19 are strongly associated with CVD [11]. Moreover, a high prevalence of pre-existing CVD has been observed among patients with COVID-19, and these comorbidities are linked to increased mortality [12]. Importantly, SARS-CoV-2 infections have been implicated in the development of CVDs, such as arrhythmias, acute coronary syndrome (ACS), heart failure (HF), myocardial injury, and venous thromboembolism [13]. It has also been demonstrated that children with COVID-19 are susceptible to developing hyperinflammatory shock syndrome with features akin to Kawasaki disease, such as coronary vessel abnormalities and cardiac dysfunction [14]. Overall, these data indicate a bidirectional interaction between CVDs and SARS-CoV-2 infection.

MicroRNAs (miRNAs) are short (∼22–25 nucleotides) endogenous non-coding RNA molecules that regulate gene expression at the post-transcriptional level [15]. The miRNA biogenesis takes place in a two-step cleavage process catalyzed by RNAse III endonucleases, namely Drosha and Dicer. Drosha catalyzes the nuclear processing of the primary miRNAs into stem-loop precursors (pre-miRNAs) of ~60 to 70 nucleotides in size [16]. Exportin-5 facilitates the export of pre-miRNAs from the nucleus to the cytoplasm via a Ran-GTP–dependent pathway [17]. In the cytoplasm, the pre-miRNA is cleaved by Dicer, resulting in the formation of the 22–25 nucleotides-long mature miRNA [18]. The miRNA expression has a great significance during various immunological phenomena because these RNA molecules regulate the expression of several immune-related genes [15]. In addition to the immune system, specific miRNAs play an essential role in the cardiovascular system [19]. Several heart-specific miRNAs that play a vital role in maintaining cardiac balance have been associated with the development of CVDs. Moreover, some miRNAs expressed in the cardiovascular system are increased in response to acute cardiac stress and also during long-term responses to chronic injury in the cardiovascular system [20]. Interestingly, SARS-CoV-2 infections have been found to affect the expression of cardiac-specific miRNAs (see [21] and references cited therein). Additionally, recent data suggest that the alteration of miRNAs can influence the development of various CVDs, such as myocarditis, myocardial injury, thrombosis, and arrhythmias [20]. Inflammation- and cardiac myocyte-specific miRNAs are upregulated in patients with severe COVID-19. Notably, cardiac-specific miRNA profiles were able to distinguish COVID-19 patients from influenza-acute respiratory distress syndrome patients indicating a rather specific response and cardiac involvement of COVID-19 [22]. However, the correlation between SARS-CoV-2 infections and alterations of cardiac-specific miRNAs is not well understood.

It is, therefore, essential to elucidate the role of SARS-CoV-2 infection on alterations of miRNA levels related to the development of CVDs. This critical review investigates the interplay between SARS-CoV-2 infection and cardiac-specific miRNAs that could be detected in the circulation and provides a window into the possible mechanisms behind these correlations as well as areas that require further investigation. We searched the Pubmed, Scopus, and Science Direct databases from February 2020 to May 2022. “COVID-19”, “SARS-CoV-2”, “cardiovascular diseases”, “micro-RNAs”, “circulating miRNAs”, “inflammation”, and their respective Medical Subject Headings (MeSH) terms were used as keywords. We also selected studies on miRNAs linked to CVDs. Only studies (both basic and clinical) that were published in the English language were included in this study.

## 2. SARS-CoV-2 Infection and Cardiovascular Diseases

The clinical manifestations of COVID-19 may range from asymptomatic or mild respiratory symptoms to severe life-threatening respiratory and cardiovascular complications [23]. As previously mentioned, pre-existing CVD increases the morbidity and mortality of COVID-19, and it has been correlated with poor disease outcomes, while SARS-CoV-2 infection can cause de novo acute and chronic CVD [24]. Acute cardiac complications include HF, myocardial injury, arrhythmias, and myocarditis, which are significantly associated with higher in-hospital mortality [24]. It should be noted that acute cardiac injury is a common extrapulmonary manifestation of COVID-19 with potential chronic consequences [25]. The potential mechanism by which SARS-CoV-2 infection causes acute CVDs involves direct damage caused by viral invasion of cardiomyocytes and indirect damage through systemic inflammation [26].

SARS-CoV-2 contributes to cardiovascular damage through systemic inflammation and hypoxia, causing the following complications [27]: (1) cytokine storm (discussed further below), (2) myocardial injury, (3) coronary artery ischemia in the setting of underlying coronary artery disease and (4) elevated vascular thrombosis of small and large coronary arteries that could occur in the absence of coronary artery disease. Notably, a cardiac injury could also occur following global ischemia related to multiorgan failure, respiratory distress, and associated metabolic and hemodynamic abnormalities (see [28] and references cited therein). Overactivation of the inflammatory response and the subsequent cytokine storm, a phenomenon of hyper-inflammation that involves the overexpression of several cytokines, is crucial for the pathogenic mechanism of SARS-CoV-2 infection. For example, in patients with HF, chronic activation of the innate immune system has been detected with T-cell and macrophage myocardial infiltration and increased pro-inflammatory cytokine levels (i.e., TNF-α, IFN-γ, IL-1β, IL-6, IL-17, and IL-18) [29]. These pro-inflammatory cytokines are also present in viral myocarditis, and their sustained elevation is associated with HF progression. These cytokines are responsible for both compensatory cardiac hypertrophy and fibrosis in the setting of cardiac injury and induce further inflammation [30]. It has been proposed that in patients diagnosed with COVID-19, the cytokine storm contributes to increased myocardial oxygen consumption, suppressed cardiac function, and endothelial dysfunction [31]. Notably, the cytokine storm in COVID-19 patients is characterized by increased IL-6 [12]. Even though it has been previously demonstrated that inflammatory cytokines, such as TNF-α, induce cardiomyocyte death, the precise mechanism of cytokine-induced myocardial injury remains inconclusive [32].

Furthermore, inflamma-miRs, a set of miRNAs that are capable of regulating inflammation, including miR-146a, miR-126-3p, and miR-21-5, exhibit age-dependent changes in blood levels and could be used as indicators of a pro-inflammatory state [33]. These miRNAs target components of the nuclear factor κB (NF-κB) pathway, which is a central mediator of inflammation [34]. MiR-223-3p, miR-155-5p, and miR-21-5p are expressed in immune cells and are involved in the innate immune response as well as in HF and myocardial infarction (reviewed in [35]). These miRNAs regulate the production and secretion of pro-inflammatory cytokines in the heart by Toll-like (TRL) receptors and their downstream signaling pathway, which involves the transcription of NF-κB. Another set of miRNAs, namely miR-146a-5p and miR-155-5p, form a unique regulatory network for the fine-tuning of the macrophage inflammatory response via the regulation of NF-κB activity [36].

As the inflammatory response develops, the transcription of miR-146a-5p increases, inhibiting its targets (i.e., Interleukin-1 receptor-associated kinase 1/IRAK1 and TNF receptor-associated factor 6/TRAF6), and thus activation of NF-κB is reduced. Moreover, miR-21-5p targets programmed cell death 4 (PDCD4), a component of the NF-κB pathway, thus stimulating the production of pro-inflammatory cytokines and inhibiting the production of anti-inflammatory cytokines [37]. Also, miR-155-5p contributes to myocardial damage through the modulation of monocyte-macrophages cardiac infiltration and T-lymphocyte activation in both human and mouse viral myocarditis [35]. Alterations in the levels of a set of miRNAs, which are heart- and muscle-specific, also named myomiRs (miR-1-3p, miR-133a-3p, miR-133b-3p, miR-208a/b-3p, and miR-499-5p) have also been implicated in the development of CVDs (reviewed in [35]).

## 3. Are Circulating miRNAs a Linker between SARS-CoV-2 Infection and Cardiovascular Diseases?

To gain a deeper understanding of the relationship between SARS-CoV-2 infection and CVD, as well as the role of circulating miRNAs in the development of both diseases, it is essential to investigate the fundamental biological mechanisms involved in viral entry into host cells, the immune response, and organ injury. As previously mentioned, SARS-CoV-2 uses the ACE2 transmembrane protein that is highly expressed in the heart, lungs, kidneys, and gut to gain entry into the host cells [38]. A recent single-cell RNA sequencing study showed that more than 7.5% of myocardial cells express the ACE2 receptor protein, which could mediate SARS-CoV-2 entry into cardiomyocytes and, in this way, cause direct cardiotoxicity [39]. Therefore, SARS-CoV-2 can induce direct cardiotoxicity by infecting cardiomyocytes and developing myocarditis [28]. Viral particles have also been identified within the cardiac tissue of several patients [40].

Furthermore, a recent single-cell RNA-seq analysis of the heart revealed low levels of ACE2 in cardiac myocytes and elevated levels in pericytes, while it has been demonstrated that ACE2 was upregulated in failing hearts [41,42]. As aforementioned, ACE2 is also expressed in endothelial cells of several organs [6,43,44], and the infection of these cells by SARS-CoV-2 could be vital in vascular events that have been observed in COVID-19 patients [43]. Furthermore, due to its large size (80–100 nm), SARS-CoV-2 may need to infect endothelial cells to infect organs, including the heart or kidney. It has also been reported that SARS-CoV-2 infects engineered human blood vessel organoids derived from human induced pluripotent stem cells (iPSCs) [44]. Infection of the blood vessel organelle was inhibited by a human soluble recombinant ACE2 [44].

There is limited research on the connection between SARS-CoV-2 and miRNA deregulation, but some recent studies have explored this relationship. Some miRNAs which are encoded by RNA viruses may be involved in the regulation of viral and/or host gene expression to promote favorable conditions, which lead to viral replication [45]. Several hospitalized COVID-19 patients exhibit a systemic dysregulation of pro-inflammatory cytokines, a phenomenon known as a cytokine storm [46]. The overproduction of early response pro-inflammatory cytokines (TNF-α, IL-6, and IL-1β) leads to an increased risk of vascular hyperpermeability, multiorgan failure, and eventually death when the high cytokine concentrations are unabated over time [47]. The immune hyperinflammatory response, mediated by dysregulated macrophages and innate and adaptive immunity, has been considered a common condition in severe COVID-19 cases [48].

Investigating miRNAs that are involved in both SARS-CoV-2 and cardiovascular pathology could contribute to understanding the relationship and the positive correlation between COVID-19 and increased susceptibility to CVDs. However, due to the recent outbreak of the COVID-19 pandemic, only a limited number of studies focusing on elucidating the dual role of miRNAs in SARS-CoV-2 infection and CVDs have been published. Alteration of miRNA expression levels (either increase or decrease) is linked with the disturbance of normal physiological functions and contributes to numerous diseases, including CVDs [27]. For example, it has been reported that levels of miRNA-16 were elevated in the peripheral blood of patients with COVID-19, and higher levels of miRNA-16 were associated with more severe disease [49]. Interestingly, Wicik et al. [50] found that miRNA-16 may be involved in the interactions between the SARS-CoV-2 and the functional networks of ACE-2, which plays a vital role in the development of acute myocardial infarction (AMI). While the exact ways in which miRNAs contribute to viral infections are not fully understood, changes in the levels of specific miRNAs after SARS-CoV-2 infection have raised the possibility that they may be involved in the development of COVID-19 (discussed further below).

Overall, at the time of the publication of this work, only a limited number of miRNAs (miR-146a, miR-27/miR-27a-5p, miR-451, miR-486-5p, miR-21, miR-155, and miR-133a) that have been implicated in the development of CVDs were identified to be altered following infection with SARS-CoV-2 (Table 1). However, the exact roles of these miRNAs remain inconclusive, and further studies are required to fully understand their mechanisms of action and potential connections to CVDs.

The potential roles of these miRNAs in the development of both CVDs and COVID-19 are discussed in the following paragraphs.

### 3.1. MiR-146a

The inflammatory response to SARS-CoV-2 infection commonly results in marked activation of coagulation—the process of thromboinflammation—with evidence of systemic endothelial damage and a resultant loss of normal anticoagulant properties [69,70]. Furthermore, it has been demonstrated that COVID-19 patients with severe symptoms are characterized by hyper-inflammatory responses due to dysregulation of pro-inflammatory cytokines, resulting in excessive endothelial cell and lung damage [71]. MiRNAs regulate inflammatory responses, and miR-21-5p, miR-146a, and miR-126-3p act as pro-inflammatory state markers by targeting components of the NF-κB pathway [54]. MiR-146a is particularly noteworthy among the miRNAs that have been linked to inflammation. In immune cells, miR-146a and miR-155 are the first miRNAs that are induced after immune activation. These miRNAs modulate the TLR-signaling pathway, which triggers the production of a large variety of inflammatory cytokines and specifically IL6, Type I interferons (IFNs), and antiviral proteins through the NF-ĸB pathway [72]. Even though miR-146a-5p does not target the mRNAs encoding IL-6 or its receptor, elevated levels of serum IL-6 have been detected in miR-146a-5p^-^/^-^ mice indicating a functional link between miR-146a-5p and IL-6 levels [73]. The transcription of miR-146a-5p is also under the control of NF-κΒ despite IL-6 and miR-146a-5p playing opposite roles in the inflammatory process [74]. Interestingly, in-silico analyses show that miR-146 and -155 negatively regulate genes encoding SARS-CoV-2 cellular entry factors and their interactors are present in the oral cavity [75]. However, the role of miRNA-146a in the development of CVDs following infection with SARS-CoV-2 remains inconclusive.

Sabbatinelli et al. [54] evaluated the levels of circulating miR-146a, miR-21-5p, and miR126-3p in COVID-19 patients that were treated with the monoclonal antibody Tocilizumab (TCZ), an IL-6 signaling inhibitor, to evaluate the role of these pro-inflammatory-linked miRNAs. It has been previously demonstrated that the levels of IL-6 correlate with COVID-19 severity [76], while miR-146a-5p is a negative regulator of NF-κΒ, which is in turn a transcription factor of the gene encoding IL-6 [77]. Interestingly, Sabbatinelli et al. [54] reported an inverse relationship between miR-146a and IL-6. In detail, this study revealed increased levels of IL-6 and reduced levels of miR-146a-5p in COVID-19 patients compared to healthy age-matched control subjects, highlighting the role of the imbalance between IL-6 and miR-146a-5p in the pathogenesis of COVID-19. Furthermore, Sabbatinelli et al. [54] reported that COVID-19 patients responding to TCZ expressed higher levels of miR-146a compared to patients who did not respond to TCZ. Interestingly, COVID-19 severity was higher in non-responders with the lowest levels of miR-146a-5p. Together, this study demonstrated that in COVID-19 patients, the serum levels of miR-146a are altered, and this miRNA is a valuable indicator of the clinical course of COVID-19 response to treatment against hyper-inflammatory states. It has been suggested that the downregulation of circulating miR-146a observed in hypertension, diabetes, and obesity may explain the more severe COVID-19 cases occurring in these patients [78]. Therefore, it could be suggested that COVID-19 patients with downregulated miR-146a expression are expected to have elevated levels of pro-inflammatory cytokines, specifically IL-6, and increased susceptibility to cytokine storms. The role of IL-6 in COVID-19 progression has been also highlighted by Vasuri et al. [57], who reported that IL-6 levels were higher in COVID-19 patients compared to healthy controls.

Gao et al. [79] showed that miR-146a weakens sepsis-induced cardiac dysfunction. In another study, Oh et al. [52] reported that miR-146a was upregulated in failing cardiomyocytes and that overexpression of this miRNA suppresses the expression level of small ubiquitin-like modifier 1 (SUMO1), which reduces sarco-endoplasmic reticulum calcium ATPase-2 (SERCA-2) sumoylation leading to cardiac contractile dysfunction. On the contrary, Huang et al. [53] demonstrated that the expression levels of miR-146a were elevated in peripheral blood and correlated with IL-6 and TNF-α expression. This correlation is also proportional to plaque vulnerability and the degree of stenosis in carotid atherosclerosis. Furthermore, it has been demonstrated that miR-146a has a cardiomyocyte-protective function, and it increases cardiomyocyte viability and protects against oxidative stress [80]. Together, the above data suggest that miR-146 expression may be inversely correlated between the two conditions, i.e., CVDs and COVID-19. Even though the mechanism by which miR-146a regulates the expression of pro-inflammatory cytokines remains unclear, there is a strong link between miR-146a and hyper-inflammatory processes observed in both SARS-CoV-2 and atherosclerosis. Therefore, disruption of the miR-146a/IL6 balance might impair the modulation of pro-inflammatory cytokines and facilitate their secretion to induce a hyper-inflammatory state. It has been proposed that the downregulation of miR-146a might be one of the factors underlying severe COVID-19 due to an insufficient host antiviral response, showing more prolonged and excessive cytokine release and lack of a feedback mechanism to limit inflammatory damage to tissues [78]. Overall, it can be assumed that there are distinct alteration patterns of miR-146a in different pathological conditions and that alterations in miRNA-146a following infection with SARS-CoV-2 may contribute to the development of CVDs, including atherosclerosis.

### 3.2. MiR-27

It has been demonstrated that the levels of the miR-27 family (including miR-27a-5p) are altered in patients with cardiac remodeling and atherosclerosis, while miR-27 is overexpressed in patients with coronary artery disease [55,56]. In detail, miR-27 is linked with cardiac remodeling and the formation of atherosclerotic plaques through the activation of macrophages participating in tissue fibrosis and by positively regulating the synthesis of low-density lipoprotein (LDL) and cholesterol [55]. The upregulation of miR-27a is also linked with promoting angiogenesis, a vital adaptation mechanism in ischemia-reperfusion [81]. Interestingly, miR-27 is downregulated in patients with SARS-CoV-2 [57,58], while it has been suggested that this is a negative regulator of the IL-6 expression. More specifically, in the study by Vasulri et al. [57], an analysis of four atherosclerotic and normal femoral arteries from a COVID-19 patient with several cardiovascular comorbidities that developed pneumonia and bilateral interstitial lesions revealed a significant decline in miR-27a-5p [57]. The study also showed an increase in both the mRNA and the IL-6 protein expression, a hyperinflammatory status, and a loss of endothelial cell homeostasis and suggested that perivascular endothelialitis is linked with miR-27a-5p. The association of IL-6 expression with miR-27a-5p in this study emphasizes an essential contribution of this miRNA in SARS-CoV-2 pathology.

These results suggest an inverse relationship between the expression levels of miR-27 in the two pathological conditions, i.e., CVDs and COVID-19, suggesting that miR-27 does not contribute to atherosclerosis following infection with SARS-CoV-2. However, the precise role of miR-27 in the pathophysiological mechanisms underlying CVDs was not evaluated in the obtained articles, and no research study has found reduced miR-27 expression levels in CVD patients.

### 3.3. MiR-21 and miR- 155

In a recent study, Garg et al. [22] compared the levels of five cardiac-specific circulating miRNAs, including miR-21 and miR-126 (associated with cardiac fibroblast and endothelial cell dysfunction), miR-155 (linked to inflammation), and miR-208a and miR-499 (related to myocardial/cardiomyocyte damage) in COVID-19 patients, influenza-associated acute respiratory distress syndrome (Influenza-ARDS) admitted to the intensive care unit and healthy individuals. The serum concentration of inflammatory miR-155, heart muscle-specific miR-208a, and miR-499, as well as that of the fibromiR miR-21, were significantly increased in COVID-19 patients compared to healthy controls. Furthermore, the levels of miR-21, which is associated with fibrosis [82], were elevated in acute COVID-19 patients compared to Influenza-ARDS patients and healthy controls. Notably, the altered levels of miR-155 and miR-499 could further differentiate COVID-19 patients from Influenza-ARDS patients, despite both diseases being very similar phenotypically, indicating a rather specific response and cardiac involvement of COVID-19. MiR-21 regulates cardiac structure and function by regulating the ERK–MAP kinase signaling pathway in cardiac fibroblasts, and it is increased during heart failure, where it acts as a promoter of interstitial fibrosis and cardiac dysfunction [82]; i.e., its upregulation facilitates the survival of fibroblasts causing fibrosis and hypertrophy resulting in cardiac dysfunction [83].

HF has been observed in 23% of patients with COVID-19, and the regulatory pathways in which miRNAs participate, such as miR-155, may increase the risk of HF. MiR-155 is overexpressed in patients with COVID-19 and has been correlated with cardiovascular damage fibroblast proliferation, endothelial inflammation, promotion of apoptosis, and cardiomyocyte pyroptosis. These changes induce hypertrophy and ventricular dysfunction, which lead to HF. Furthermore, miR-15b is upregulated in patients with severe COVID-19, and this was associated with a significantly higher incidence of arrhythmias, indicating that this miRNA could be used as a possible marker of heart damage (see [27] and references cited therein).

### 3.4. MiR-133a

Gutmann et al. [68] examined the association of circulating miRNAs with COVID-19 severity and 28-day intensive care unit (ICU) mortality. The results of this study revealed that a total of 60 miRNAs, including cardiomyocyte- platelet-, endothelial-, and hepatocyte-derived miRNAs, were differentially expressed depending on the severity of the disease. Interestingly, the myocyte-derived (MyomiR) miR-133a [84] and cardiomyocyte-derived miR-208b [85] were detectable in patients with severe disease. Furthermore, miR-133a, which reflects inflammation-induced myocyte damage, was related to 28-day mortality and negatively correlated with neutrophil counts and proteins related to neutrophil degranulation, such as myeloperoxidase [68]. MiR-133a levels were also the highest in critically ill patients with cardiopulmonary diseases. It is worth mentioning that miR-133 participates in the proliferation, differentiation, survival, hypertrophic growth, and electrical conduction of cardiac cells, which are essential for cardiac fibrosis, cardiac hypertrophy, and arrhythmia [86]. Furthermore, myocardial injury [87] and chronic obstructive pulmonary disease (COPD) [88] are known to increase circulating miR-133a levels. Neutrophil degranulation and extravasation resulting in myocyte damage is a likely cause for the rise of circulating miR-133a [89]. Moreover, there is evidence that neutrophils may be a secondary source of miR-133a in circulation [90].

### 3.5. MiR-486-5p

MiR-486 is expressed mainly in the heart and muscle and targets proteins in cardiomyocyte survival (PI3K/PTEN/pAKT), myogenesis (MyoD), myotube survival/differentiation (DOCK3/PTEN/AKT), and cardiac progenitor cell proliferation networks. miR-486 regulates myocardial homeostasis by negatively regulating PIM1 kinase, an essential component of cardiac progenitor cell cyclin (see [91] and references cited therein). MiR-486-5p is linked with reduced cell proliferation and promotes cell death by accumulating superoxide anions and causing DNA damage. These mechanisms are tightly associated with heart failure, and cardiac remodeling as their characteristic features are increased inflammatory responses, cardiomyocyte apoptosis or necrosis, and tissue fibrosis. miR-486 also alters the activation and differentiation of B- and T-lymphocytes, prevents resolution at inflammatory sites, and downregulates the expression of the anti-inflammatory cytokine IL-10, highlighting the pro-inflammatory role of miR-486-5p [27]. It has been proposed that reduced miR-486-5p expression levels contribute to weakening the immune system and intensifying the severity of the disease by favoring viral replication [27]. Interestingly, miR-486–5p is one of the most downregulated miRNAs in lung tumor tissues and contributes to lung cancer progression [92]. The role of miR-486-5p is unclear, as it is downregulated in COVID-19 patients and upregulated in patients with atrial fibrillation, implying that COVID-19 cannot induce arrhythmias by altering miR-486-5p expression levels. The association of miR-486-5p with arrhythmias is further supported by Li and coworkers [63], whose study showed that miR-486-5p is overexpressed in patients with arrhythmias and is involved in depressing the sinoatrial node function. Although miR-486-5p upregulation in patients with arrhythmias can be associated with prolonged inflammatory responses, this highlights the notion that CVD patients with upregulated miR-486-5p can be susceptible to severe COVID-19 states characterized by hyper-inflammatory state or cytokine storm. It has been demonstrated that miR-486 is significantly downregulated in the heart upon cardiac ischemia/reperfusion (I/R) injury [64].

### 3.6. MiR-451

MiR-451 is expressed in multiple systems, including the urinary, respiratory, and digestive systems, and regulates various physiological and pathological processes, including hematopoiesis, epithelial cell polarity, and embryogenesis [93]. MiR-451 is reported to be upregulated in both the right and left atria of patients diagnosed with atrial fibrillation. One of the multiple functions of miR-451 is to prevent cell death from ischemia-reperfusion injury by mediating the cyclooxygenase-2 pathway, which is heavily linked with arrhythmias [60]. Although the role of this miRNA in atrial fibrillation can be regarded as an adaptation mechanism, the exact mechanism of action remains unknown. miR-451 was found to be downregulated in COVID-19 patients and is associated with severe outcomes, presumably by preventing the expression of pro-inflammatory cytokines, similar to miR-486-5p, and favoring viral replication [58]. The association of miR-451 with the protection against apoptosis and the strong link with severe COVID-19 phenotypes suggests a decline in miR-451 that COVID-19 patients can also increase the susceptibility to CVDs, presumably due to the lack of protection against apoptosis. Aberrant apoptotic mechanisms can contribute to extensive tissue damage, fibrosis, and cardiac remodeling [94]. Yang et al. [61] reported that compared to healthy donors, COVID-19 patients had significantly higher mRNA expression of IL-6R while miR-451a, a known negative regulator of IL-6R translation, was down-regulated. This may lead to increased IL-6 protein expression.

## 4. Conclusions

Identifying correlations between CVDs and COVID-19 is of particular interest because individuals suffering from CVDs may be at higher risk for severe illness or death from COVID-19. Recent evidence suggests that SARS-CoV-2 infection can severely damage the cardiovascular system. In a recent review, Izzo et al. [95] investigated the cardiovascular effects of miRNAs in COVID-19. They identified six cardio-specific miRNAs (miR-21-5p, miR-153, miR-155-5p, miR-214, miR-208a-3p, and miR-375) that are upregulated and nine cardio-specific miRNAs (miR-26B, miR30e-3p, miR-125b, miR-133a, miR-146a, mir-200c-3p, miR-223-3p, miR-500, and miR-590-3p) that are downregulated in COVID-19 patients. These miRNAs have been linked to various forms of cardiovascular damage, including acute ischemic injury, nonischemic injury, stress cardiomyopathy, heart failure, myocardial fibrosis, thromboembolism, vascular inflammation, and Kawasaki disease (see [95] and reference cited therein). Our critical review also indicates that alteration of the expression levels of some COVID-19-related miRNAs are correlated with CVDs and are linked with various mechanisms (Figure 1). However, how an infection with SARS-CoV-2 affects the profile and/or expression levels of cardiac-specific miRNAs, and what are the exact mechanisms by which the alteration of these miRNAs in response to SARS-CoV-2 infection leads to CVDs or worsening pre-existing CVDs remain inconclusive. Similar to Izzo et al. [95], our work highlights the potential implication of miR-21, miR-155, 133a, and miR-146a in the pathophysiology of both CVDs and COVID-19. Furthermore, in our work, miR-27/miR27a-5p, miR-451, miR-486-5p, miR-21, and miR-155 that have been implicated in the development of CVDs were identified to be altered following infection with SARS-CoV-2 (Table 1). Despite the alterations in the expression levels of these miRNAs, the identified changes and contributions to cellular processes do not correlate clearly between SARS-CoV-2 infections and cardiovascular pathologies. The results of this work also highlight the central role of IL-6 in the pathogenesis of both CVDs and COVID-19. This work emphasizes the importance of conducting further research to understand the relationship between CVDs and COVID-19 in order to develop strategies for preventing and managing COVID-19 in individuals with CVDs. This could involve studying the relationship between cardio-specific miRNAs and immune function, inflammation, or other biological pathways that may be involved in the severity of COVID-19. To effectively identify correlations between miRNAs, CVDs, and COVID-19, it will be necessary to use systematic and collaborative approaches involving basic researchers, clinical investigators, industry partners, and governmental agencies. This will help to translate biomarkers into clinical use.

## Figures and Tables

**Figure 1 ijms-24-02380-f001:**
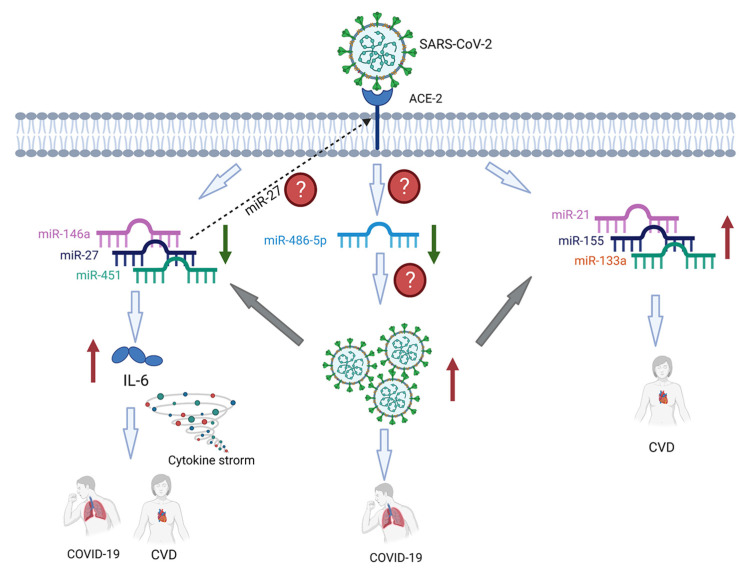
Potential associations between alterations of miRNA levels after infection with SARs-CoV-2 virus and the development of CVDs. Viral infection causes downregulation of miR-146a, miR-27, and miR-451, which are well-established inhibitors of IL-6 protein and IL-6 mRNAs, leading to the overproduction of several pro-inflammatory cytokines, a process known as cytokine storm. This phenomenon has been implicated in the pathogenesis of coronaviruses disease-19 (COVID-19) and cardiovascular diseases (CVDs). Expression levels of miR-27 could also correlate with the expression of the ACE-2 receptor, which provides the entry point of SARS-CoV-2. On the other hand, SARS-CoV-2 infection causes the upregulation of miR-21, miR-155, and miR-133a which are implicated in the development of CVD. A SARS-CoV-2 infection causes downregulation of miR-486-5p, facilitating the replication of the virus by weakening the immune system response. Question marks (?) represent factors that should be examined further.

**Table 1 ijms-24-02380-t001:** Circulating miRNAs associated with COVID-19 and cardiovascular diseases.

miRNA	Function/Alteration in CVDs	Ref	Function/Alteration in COVID-19	Ref
miRNA-146a	Altered in CVDs Promotes formation of atherosclerotic plaques through upregulation of IL-6 and TNF-α	[51,52]	Downregulated in COVID-19 patients leading to upregulation of IL6 Correlated with the severity of the disease	[53,54]
miR-27 miR-27a-5p	Overexpressed in patients with coronary artery disease Promotes fibrosis, activation of macrophages	[55,56,57]	Downregulated in COVID-19 patients leading to upregulation of IL6	[58]
			ACE2 levels are positively correlated with miR-27a/b	[59]
miR-451	Mediates the cycloocygenase-2 pathway and prevents myocardial cell death from ischemia-reperfusion injury	[60]	Downregulated in COVID-19 patients leading to overexpression of IL 6	[58,61]
miR-486-5p	Reduces cell proliferation Associated with arrhythmias	[62]	Downregulated in COVID-19 patients favoring viral replication by weakening the immune system	[27]
	Upregulated in patients with arrhythmias	[63]		
	Downregulated in the heart upon cardiac ischemia/reperfusion	[64]		
miR-21	Has been implicated in cardiac dysfunctions, including cardiac hypertrophy and fibrosis.	[65]	Elevated levels in COVID-19 patients	[22]
	Associated with the pro-inflammatory state.	[66]		
	Regulates cardiac structure Upregulated during heart failure	[22]		
miR-155	Correlated with cardiovascular damage, endothelial inflammation, fibroblast proliferation Upregulated in CVDs	[22]	Elevated levels in COVID-19 patients	[22]
miR-133a	Upregulated in patients with cardiopulmonary diseases	[67]	Elevated levels in COVID-19 patients. Associated with COVID-19 severity and 28 days mortality	[68]

## Data Availability

Not applicable.

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
