# Peer review of "Alterations in Circulating miRNA Levels after Infection with SARS-CoV-2 Could Contribute to the Development of Cardiovascular Diseases: What We Know So Far"

_ijms, 2023, doi:10.3390/ijms24032380_

Round 1

Reviewer 1 Report

The authors have researched a new growing hot topic of CVD in a review that is well structured and set to an audience of academic cardiologists. Since the pandemic many efforts have been spent to further understand the implication of COVID-19 with CVDs. A part of these efforts has investigated miRNAs involvement, highlighting interesting scientific evidence. 
The article requires moderate English adjustments and there are common punctuation errors (see chapter 3 and the miRNAs following are 2.1 as they should be 3.1 etc.).

My main concern however is a missing link between chapter 3 and the conclusion. Since the authors talk about CVDs, it is fit to identify the main ones (Heart failure, VTE, Myocarditis, arterial thrombotic events, Takotsubo syndrome) with relative miRNA association. This has been done recently by other authors, you can take inspiration from them (35282650; 35779946; 35167732). This missing aspect should also be completed with a summarizing figure (you can also modify the existing one, expanding it with the required CVDs). 

Lastly, the Conclusion is missing a future prospective piece where the authors suggest what future research should embrace and towards. 

Reviewer 2 Report

In this review Pieri et al studied the role of SARS-CoV-2 infection on alterations of miRNAs levels related to the development of cardiovascular diseases.

The topic of the study is relevant but still preliminary in this field, since may 2022 many other papers have been published in this topic. There are already some comments that should be addressed:

-        This sentence is confusing and should be rephrased (lines 76-79): “A variety of heart-specific miRNAs regulate cardiac homeostasis and 76 have been implicated in the development developing over, several miRNAs expressed in 77 the cardiovascular system are upregulated following cardiac stress but also during long-78 term responses of the cardiovascular system to chronic injury”

-        Line 93 “February 2000 to May 2022” is the date correct?

-        Line 138: “hF” abbreviation should be corrected

-        Line 224: miR-145a-5p, should it be replaced by miR-146a-5p?

Round 2

Reviewer 1 Report

The authors have extensively modified the review according to what has been requested. The issues and the points have been addressed in an according manner. The article is now fit for publication.